# Healthcare provider-to-patient perspectives on the uptake of teleconsultation services in the Nigerian healthcare system during the COVID-19 pandemic era

Ifeanyi Jude Ezeonwumelu[1]☯*, Ifeanyi Jude Obijiaku[2], Chukwudi Martin Ogbueche[3], Reunlearn360 2020 Cohort¶, Ucheoma Nwaozuru[4]☯*

1 AIDS Research Institute-IrsiCaixa and Health Research Institute Germans Trias i Pujol (IGTP), Hospital Germans Trias i Pujol, Universitat Autònoma de Barcelona, Badalona, Spain, 2 Department of Medicine, University of Benin Teaching Hospital, Edo State, Nigeria, 3 Department of Physiotherapy, Glasgow Caledonian University, Glasgow, Scotland, 4 Department of Implementation Science, Wake Forest School of Medicine, Winston-Salem, NC, United States of America

☯ These authors contributed equally to this work.
¶ Membership of Reunlearn360 2020 cohort is provided in the acknowledgments.
* iezeonwumelu@irsicaixa.es (IJE); unwaozur@wakehealth.edu (UN)

## Abstract

The urgency to curtail the devastating effects of the ongoing COVID-19 Pandemic has led to the implementation of several measures to limit its spread, including movement restrictions and social distancing. As most developing countries rely solely on hospital visitations for their medical needs, this impediment to assessing healthcare services compounded by low uptake of telehealth services could result in dire consequences. This is a cross-sectional study among Healthcare providers (HCP) and Healthcare consumers (HCC) in Nigeria. We administered a pre-validated self-administered online questionnaire comprising questions to assess the knowledge, use, perceptions, and benefits of telemedicine among study participants. Descriptive statistics were used to examine participants' perceptions on telemedicine use and to summarize participants' characteristics. A total of 158 healthcare providers and 1381 healthcare consumers completed the online survey. Ninety percent of HCP reported that they used some form of telemedicine to deliver health care, and 63% of HCC had received healthcare through telemedicine. A significant proportion of HCP (62%) and HCC (69%) agreed that telemedicine would improve healthcare consultation experience and satisfaction. However, fewer (21%) HCP agreed that they liked that there would be no physical contact with patients using telemedicine. In contrast, 52% of HCC agreed that they liked that there would be no physical contact with healthcare providers while using telemedicine. The majority of the participants believed that benefits of telemedicine would include: being a safe way for healthcare delivery during pandemics (HCP = 62%, HCC = 83%), affordability (HCP = 62%, HCC = 82%), and time-saving (HCP = 54%, HCC = 82%,). Teleconsultation services have been shown to aptly complement face-to-face hospital visits in ensuring effective triaging in hospitals and providing adequate healthcare delivery to patients regardless of geographical and physical barriers. These results support telemedicine use for the provision of healthcare services during the COVID-19 pandemic.

**Data Availability Statement:** All relevant data are within the manuscript and supplementary information.

**Funding:** The authors received no specific funding for this work.

**Competing interests:** The authors have declared that no competing interests exist.

## Introduction

The ongoing COVID-19 Pandemic has necessitated the urgent and crucial need to adopt strategies in the healthcare sector that seek to minimize face-to-face consultations, triage hospital visitations, and opt for teleconsultations whenever possible to limit the spread of the SARS--CoV-2 virus [1, 2]. The adoption of telehealth services during this Pandemic has ensured the continuity of healthcare delivery–breaking down physical and geographical barriers to assessing medical care even as most countries worldwide enforced movement restrictions and physical distancing measures aimed at reducing community nosocomial spread [3, 4].

Telehealth refers to the delivery of health care services from healthcare providers (HCP) to patients separated by physical (geographical) distance using information communication technology (ICT) for the diagnosis and treatment of diseases and injuries, research, and evaluation. Telehealth services (teleconsultations), including HCP-to-patient and HCP-to-HCP communications, can be delivered synchronously (real-time telephone and video calls), asynchronously (patient portals and e-consults), and through virtual agents (chatbots) [5, 6]. Now, the world must have to adapt to this recent Pandemic; it is time we re-evaluated the emerging role of telehealth as a key player in healthcare service delivery and possibly, revisit the question recently posed by Duffy and Lee—whether face-to-face visits should become the second, third, or even last option rather than the first option for meeting routine patient needs [5].

Although the recent COVID-19 Pandemic has changed world economies and healthcare systems; on the contrary, it could have a knock-on effect on the rapid uptake of telehealth services [7, 8]. However, there is still a gap in the adoption of telehealth services between developed countries with robust and efficient healthcare facilities and developing countries with less efficient healthcare systems, which has become more evident during this Pandemic [9]. The projection of Africa as the next epicenter of the ongoing COVID-19 Pandemic is indeed worrisome given the devastation this Pandemic could cause to its vulnerable healthcare systems [10]. The region's healthcare system is faced with limited health care facilities and health provider shortages [11]. Hence, as COVID-19 cases in Nigeria continues to surge [12], the increasing burden of this Pandemic to its fragile healthcare system and resources necessitates proactive measures to rescue its healthcare system from eventual collapse.

The uptake of teleconsultation services in the Nigerian healthcare system is yet to gain traction in recent times due to several challenges, including poor telecommunication infrastructure, socio-cultural influence, change resistance, lack of awareness, fear of privacy and confidentiality breaches, and impersonation due to quackery, medico-legal issues and willingness to pay for telehealth services [13, 14]. However, smartphone penetration in Nigeria has been increasing over the years, with over 196 million and 143 million active telephone and internet subscribers recorded in June 2020, corresponding to an increase of 12.6% and 17.2%, respectively, from a comparable period the previous year [15]. This trend is promising and could be leveraged to improve the teleconsultation infrastructure in Nigeria.

This study aimed to evaluate the perception of the acceptability and usefulness of teleconsultations in the Nigerian healthcare system from both the HCP and healthcare consumers' (HCC) perspectives and determine the influential role of the COVID-19 Pandemic on the uptake of teleconsultation services in Nigeria.

## Methods

This study was an observational, cross-sectional study using an online, self-administered questionnaire.

## Study participants, sample size, and sampling

The Open-Source Epidemiologic Statistics for Public Health (OpenEpi), v.3.01 (updated 2013/04/06) was used to calculate the sample size for this survey. Here, we hypothesized that at a 95% confidence level, 50% of the respondents must have been exposed to teleconsultations and are twice as likely to have positive acceptability and perception towards the uptake of teleconsultation services. The required sample size was estimated to be 1034 respondents, and with an additional 30% contingency, 1345 respondents from the public were assumed as an appropriate sample size. Assuming an HCP: Patient ratio of 1:10 in Nigeria, an additional 135 respondents from the HCP population were targeted for the HCP survey. Data collection was performed online using the google forms survey tool (Alphabet Inc., California, USA). The call for participation was made on social media (WhatsApp, Telegram, Facebook, LinkedIn, and Twitter) and was open for 3 months or until the required sample size was reached (02/11/2020-31/12/2020). Participation in this survey was voluntary, anonymous and participants could exit the survey at any time.

## Research instrument

The research instrument for this study includes two online self-developed questionnaires. Two sets of survey tools (S1 and S2 Surveys), one for each of the target respondent groups–HCP and HCC and was pre-validated by at least three independent reviewers followed by a pre-test survey of 200 respondents from the outpatient clinics at University of Benin Teaching Hospital (UBTH). The questionnaire consists of five sections: a) a consent and approval page, b) demography of respondents, c) healthcare background check, d) acceptability and usability of telephone-delivered medical consultations, e) teleconsultations during the COVID-19 Pandemic. Sections 4 and 5 of the questionnaires were structured as positively framed statements, with respondents being asked to rate their agreement with each statement on a 5-point Likert scale ranging from strongly agree to disagree strongly. In addition, to ascertain the willingness to pay for teleconsultations, respondents were asked to respond to a statement regarding an appropriate fee to pay for teleconsultations rated on a 5-point Likert scale, ranging from "50% more than the cost of a face-to-face hospital visitation" to "50% less than the cost of a face-to-face hospital visitation."

## Data analysis

Data were summarized using Microsoft Excel 2019 and analyzed utilizing the Statistical Package for the Social Sciences (SPSS) software, v.22, and the OpenEpi. Data were analyzed using descriptive statistics and frequencies to examine the characteristics of study participants and their responses to the survey items. Data are presented as frequencies and proportions. Bivariate analyses were conducted to examine whether there were differences in participants' responses to overall perceptions on telemedicine and benefits of telemedicine across participants' demographic factors. For bivariate comparison, chi-square or Fisher's exact for smaller sample groupings was used. For comparative analysis, participants perceptions were dichotomized by aggregating the categories "agree" and "strongly agree" to "agree" in the text, while "neutral", "disagree" and "strongly disagree" were combined to "disagree." Statistical significance was defined at the level of a p-value equal to or less than 0.05.

## Ethical approval

This study was approved by the health research ethics committee of the University of Benin Teaching hospital with the protocol number: ADM/E22/A/VOL.VII/14830854. Per our

study protocol, participation was completely voluntary. All questions were self-administered and anonymous. Participants were not compensated for completing the survey and were free to withdraw or not answer any questions they did not want to with no professional or other consequences. Informed consent (written) was obtained from all subjects involved in the study.

## Results

### Characteristics of healthcare providers

Of the 168 HCP who completed the response, 158 of them resided in Nigeria. The results are based on the 158 HCP who reside in Nigeria. The mean age of the HCP is 31years (S.D = 4.9). Most of the HCP were male 95(60%) and resided in the South-South region of Nigeria 67 (42%). About half (52%) of the HCP practiced exclusively in public health systems, 16% exclusively in private health settings, and 32% practiced in a combination of public health and private health settings. The majority of the HCP 115(73%) had experience with telemedicine consultation over the phone and had experience with telemedicine involving video 28(17%). A summary of HCP characteristics is provided in Table 1.

### Healthcare providers overall perceptions on telemedicine for healthcare delivery

Healthcare providers (HCP) who participated in the study expressed opinions regarding their perceptions on using telemedicine for healthcare delivery (Fig 1). The results are presented for participants who agreed or strongly agreed to the statements. Less than half of the HCP (45%, 71) indicated that they would be interested in healthcare services offering medical consultations over the phone for patients. A good number of the HCP (45%, 70) agreed or strongly agreed that they would only prefer to use teleconsultation services during pandemics such as COVID-19. Only a few participants (24%, 39) perceived that they would be as satisfied talking to patients over the phone as they would talking to a patient during an in-person consultation.

Additionally, 62% (98) agreed or strongly agreed that video over the internet services such as WhatsApp, Skype, or Zoom will improve their teleconsultation experience and satisfaction. Forty-two percent (67) of the participants expressed that using a phone to consult with a patient and prescribe medications would be an easy process. Very few (21%, 34) of the HCP liked that there would be no physical contact with patients when consulting over the phone.

In this section, we also highlight the notable differences in perception of telemedicine use by HCP demographics. The bivariate categorical test shows no relationship with HCP's age, location (geopolitical zone), gender, recent professional development status, and healthcare service delivery setting on their responses to the perception statements (whether they agreed or disagreed with perception statements). The only exception was the perception statement on whether HCP agreed or disagreed that videos would improve their teleconsultation experience and satisfaction. There was a statistically significant association between participants' geopolitical zone and their responses to the aforementioned statement ($p$-value = 0.042). Likewise, there was a statistically significant association between participants' current professional development status and their responses to the statement on if they agreed or disagreed that videos would improve their teleconsultation experience and satisfaction ($p$-value = <0.0001). A complete list of comparisons can be found in the S1 Table.

**Table 1. Healthcare providers characteristics (n = 158).**

| Variables | n (%) |
|---|---|
| Age | 30.5±4.9 (Mean ±S.D) |
| **Geopolitical Zone** | |
| Northeast | 5(3%) |
| North-West | 5(3%) |
| North Central | 7(4%) |
| South-West | 20(13%) |
| South-East | 54(34%) |
| South-south | 67(42%) |
| **Gender** | |
| Female | 63(40%) |
| Male | 95(60%) |
| **Highest Educational Qualifications** | |
| MBBS+ PhD | 1(1%) |
| MBBS+ Masters | 16(10%) |
| Other Degrees | 38(24%) |
| MBBS | 103 (65%) |
| **Most Recent Professional Development Status** | |
| Consultancy | 4(3%) |
| Residency | 37(23%) |
| Internship | 45(29%) |
| Other Trainings | 72 (46%) |
| **Healthcare Service Delivery setting** | |
| Exclusively in private health settings | 25(16%) |
| In a combination of public health and private health settings | 51(32%) |
| Exclusively in public health system | 82(52%) |
| **Frequency treating patients in hospital/healthcare facility in the last 6 months before the COVID-19 Pandemic** | |
| Infrequently, at most 1 in the last 6 months | 4(3%) |
| Somewhat frequently, between 2 and 5 times in the last 6 months | 7(4%) |
| Frequently; at least 1 per month | 8(5%) |
| None | 13(8%) |
| Very frequently; at least 1 per week | 126(80%) |
| **Frequency treating patients in hospital/healthcare facility since the onset of the COVID-19 Pandemic** | |
| Infrequently, at most 1 in the last 6 months | 5(3%) |
| None | 6(4%) |
| Somewhat frequently, between 2 and 5 times in last 6 months | 12(8%) |
| Frequently; at least 1 per month | 14(9%) |
| Very frequently; at least 1 per week | 121(77%) |
| **Experience with telemedicine consultation** | |
| Yes, via video over internet (e.g., WhatsApp, Skype) | 10(6%) |
| None | 15(10%) |
| Yes, over the Phone, Yes, via video over the internet (e.g., WhatsApp, Skype) | 18(11%) |
| Yes, over the Phone | 115(73%) |
| **A session of medical consultations over the phone should cost patients . . .** | |
| 50% more than the cost of a face-to-face hospital visitations | 10(6%) |
| 25% more than the cost of a face-to-face hospital visitations | 24(15%) |

(*Continued*)

**Table 1.** (Continued)

| Variables | n (%) |
|---|---|
| The same cost as a face-to-face hospital visitation | 37(23%) |
| 25% less than the cost of a face-to-face hospital visitations | 42(27%) |
| 50% less than the cost of a face-to-face hospital visitations | 45(29%) |

## Healthcare providers perceptions on benefits of telemedicine for healthcare delivery

The majority of the HCP expressed opinions regarding the perceived benefits of using telemedicine for healthcare delivery (Fig 2). Most of the participants, 54% (86), indicated that delivering medical consultations and prescriptions to patients over the phone would help in saving the patients' time. However, less than half (40%,64) indicated that delivering medical consultations and prescriptions to patients over the phone would be a convenient form of healthcare for patients. Sixty-four percent (102) of the participants expressed that using phones for healthcare service delivery during the COVID-19 Pandemic would be a safe strategy for patients to receive care. Additionally, sixty-one percent (96) of the participants indicated that using the phone would be an affordable way for patients to receive healthcare services during the COVID-19 Pandemic. Fifty-three percent (84) of participants either agreed or strongly agreed with the statement that using a phone would be a useful, practical, and effective way for patients to receive healthcare service from a doctor or health professional during the COVID-19 Pandemic.

In addition, there were some notable differences in perception on the benefits of telemedicine use by HCP demographics. There was a statistically significant association between participants recent professional development status (*p-value* = 0.008), healthcare service delivery setting (*p-value* = 0.033), and if participants agreed or disagreed, using the phone would be a useful (practical) and effective way for patients to receive healthcare service from a Doctor/Health professional during COVID-19 pandemic. There was no relationship between this response and participants' age, geopolitical zone, and gender. Also, participants' recent professional development status and healthcare service delivery setting were associated with whether

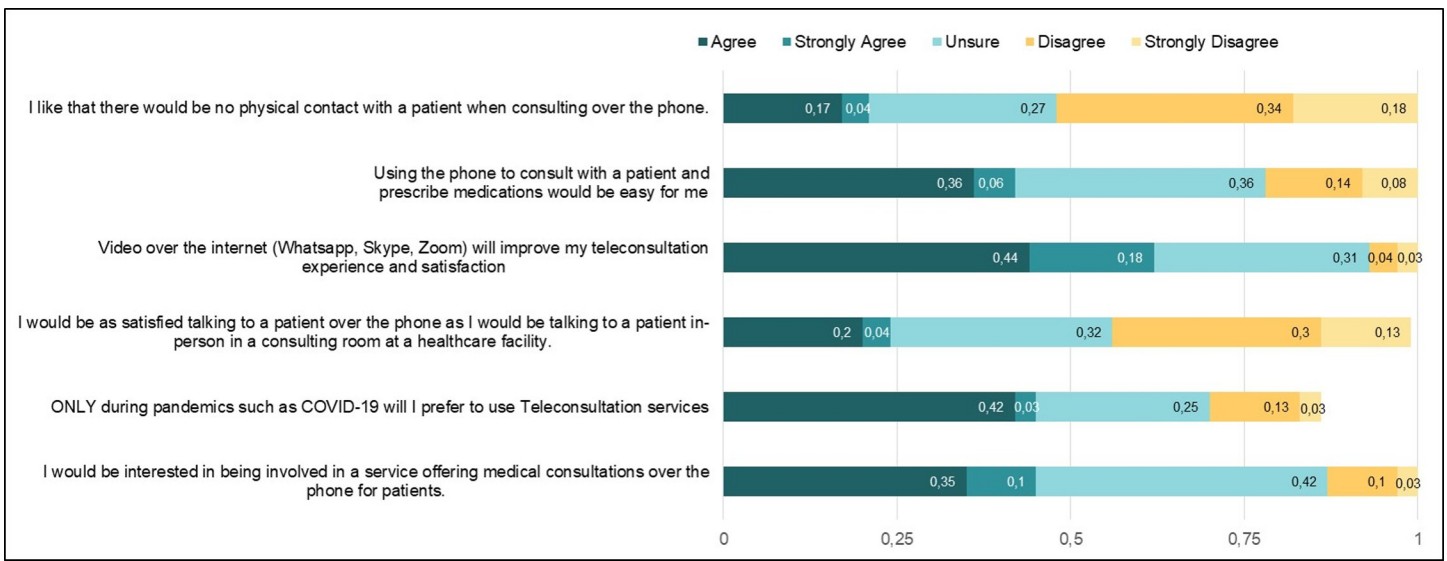

**Fig 1. Overall perceptions on the use of telemedicine among healthcare providers.**

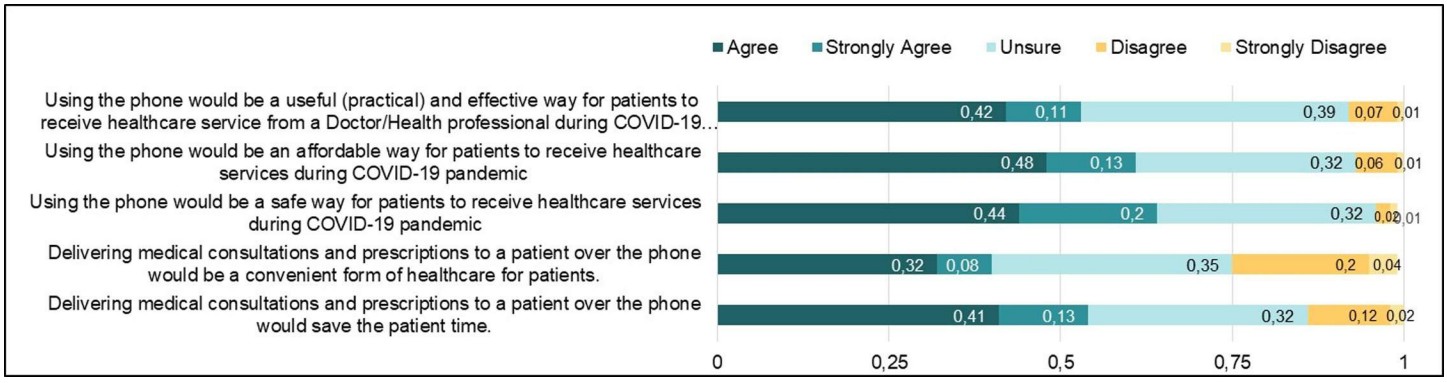

**Fig 2. Perception on the benefits of telemedicine among healthcare providers.**

they agreed or disagreed that phone would be an affordable way for patients to receive healthcare services during the COVID-19 pandemic. Complete details of results can be found in the S2 Table.

## Characteristics of healthcare consumers

A total of 1381 participants who were not physicians or HCP completed the survey. The mean age of the participants is 28.81 years (S.D = 6.3). The majority of the participants were male, 799(57.9%), and resided in the South-East region of Nigeria 560(40.6%). Most participants had a formal education with an almost equal split in non-scientific/non-medical and scientific/medical fields. The majority of the participants, 546(39.6%), visited a combination of public health and private health settings for their healthcare needs. About 42% (576) participants visited health facilities infrequently, at most once in six months preceding the COVID-19 Pandemic. Also, about half of the participants indicated that they did not experience more hospital cancellations or long-term rescheduling of appointments due to the pandemic. The majority of the participants, 510(37%), had never received telemedicine consultation during the data collection. The Summary of study participants' characteristics is provided in Table 2.

## Healthcare consumers overall perceptions on telemedicine for healthcare delivery

Healthcare consumers who participated in the study expressed their perceptions of using telemedicine for healthcare services (Fig 3). The results are presented for participants who agreed or strongly agreed to the statements. Most of the participants agreed 774 (56%) that they would be interested in healthcare services offering medical consultations over the phone for patients. Fifty-eight percent of the participants agreed 576(42%) or strongly agreed 215(16%) that they would only prefer teleconsultation services during pandemics such as COVID-19. Most of the participants perceived that they would be satisfied talking to a healthcare provider over the phone as they would during in-person consultation (agree 659(48%); strongly agree 338(21%)).

## Healthcare consumers perceptions on benefits of telemedicine for healthcare delivery

Eighty-three percent (n = 1150) of the participants expressed that using phones to receive healthcare service during the COVID-19 Pandemic would be a safe strategy. The majority of the participants indicated that using a phone would be an affordable 1135(82%) and practical

**Table 2. Healthcare consumers characteristics (N = 1381).**

| Variables | n (%) |
|---|---|
| Age | 28.81±6.3 (Mean ±S.D) |
| **Geopolitical Zone** | |
| South-East | 560(40.6) |
| South-West | 402 (29.1) |
| South-South | 282 (20.4) |
| North-Central | 90 (6.5) |
| North-West | 27 (2.0) |
| North-East | 14 (1.0) |
| **Gender** | |
| Male | 799 (57.9) |
| Female | 573 (41.9) |
| **Highest Education Level** | |
| Bachelors | 716 (51.9) |
| Secondary | 291 (21.1) |
| Masters | 283 (20.5) |
| Doctorate | 41 (3.0) |
| Others | 35 (2.5) |
| No Formal Education | 9 (0.7) |
| **Academic Background** | |
| Non-Scientific/Non-medical | 686 (49.7) |
| Scientific/Medical | 689 (49.9) |
| **Frequency in Visiting Health Facility 6 months Prior to the COVID-19 Pandemic** | |
| Infrequently; at most 1 in the last 6 months | 576 (41.7) |
| None | 315 (22.8) |
| Somewhat frequently, between 2 and 5 in the last 6 months | 182 (13.2) |
| Very frequently; at least 1 per week | 166 (12.0) |
| Frequently; at least 1 per month | 136 (9.9) |
| **How frequently did you visit a hospital/healthcare facility since the onset of the COVID-19 Pandemic?** | |
| None | 503 (36.4) |
| Infrequently, at most 1 in the last 6 months | 483 (35.0) |
| Somewhat frequently, between 2 and 5 in the last 6 months | 162 (11.7) |
| Very frequently; at least 1 per week | 147 (10.7) |
| Frequently; at least 1 per month | 80 (5.8) |
| **Had more Hospital visitation cancellations or long-term rescheduling of appointments since the COVID-19 Pandemic** | |
| No | 704 (51.0) |
| Yes | 510 (37.0) |
| Maybe | 161 (11.7) |
| **Usual Healthcare Service Location** | |
| In a combination of public health and private health settings | 546 (39.6) |
| Exclusively in private health settings | 377 (27.3) |
| Other (e.g., Pharmacy stores, Self-medication, Referrals, etc.) | 294 (21.3) |
| Exclusively in the public health system | 158 (11.4) |
| **Ever received telemedicine consultation** | |
| None | 510 (37.0) |
| Yes, over the Phone | 393 (28.4) |

(*Continued*)

**Table 2.** (Continued)

| Variables | n (%) |
|---|---|
| Yes, over the Phone, Yes, via video over the internet (e.g., WhatsApp, Skype) | 307 (22.2) |
| Yes, via video over internet (e.g., WhatsApp, Skype) | 165 (12.0) |
| **A session of medical consultations over the phone should cost** | |
| 50% less than the cost of a face-to-face hospital visitations | 480 (34.8) |
| 25% less than the cost of a face-to-face hospital visitations | 530 (38.4) |
| The same cost as a face-to-face hospital visitation | 187 (13.6) |
| 25% more than the cost of a face-to-face hospital visitations | 92 (6.7) |
| 50% more than the cost of a face-to-face hospital visitations | 86 (6.2) |

1135(82%) way to receive healthcare service during the pandemic. Also, most of the participants, 82% (n = 1135), indicated that receiving medical consultations and prescriptions over the phone would help in saving time. Additionally, 69% (n = 954) of the participants agreed that receiving medical consultations over the phone would be convenient for them (Fig 4).

## Comparison of healthcare providers and healthcare consumers responses on overall perceptions and benefits of telemedicine use

Table 3 presents the results of the bivariate comparison of HCP and HCC responses on overall perceptions of telemedicine use. There was a statistically significant association in the proportion of HCP and HCC who agreed to the overall telemedicine perception statements ($p < 0.05$). The exception was for the proportion of HCP or HCC who agreed that they prefer to use teleconsultation services ONLY during pandemics. There was no statistically significant association in the proportion for the two groups ($p = 0.497$).

In addition, Table 4 presents the results of the bivariate comparison of HCP and HCC responses on the benefits of telemedicine use. There was a statistically significant association in the proportion of HCP and HCC who agreed to the benefits of telemedicine use statements ($p < 0.05$).

## Discussion

### Main findings

Telemedicine is garnering attention given the prominence of mobile technologies in sub-Saharan Africa countries, including Nigeria [16]. The emergence of the COVID-19 pandemic has further

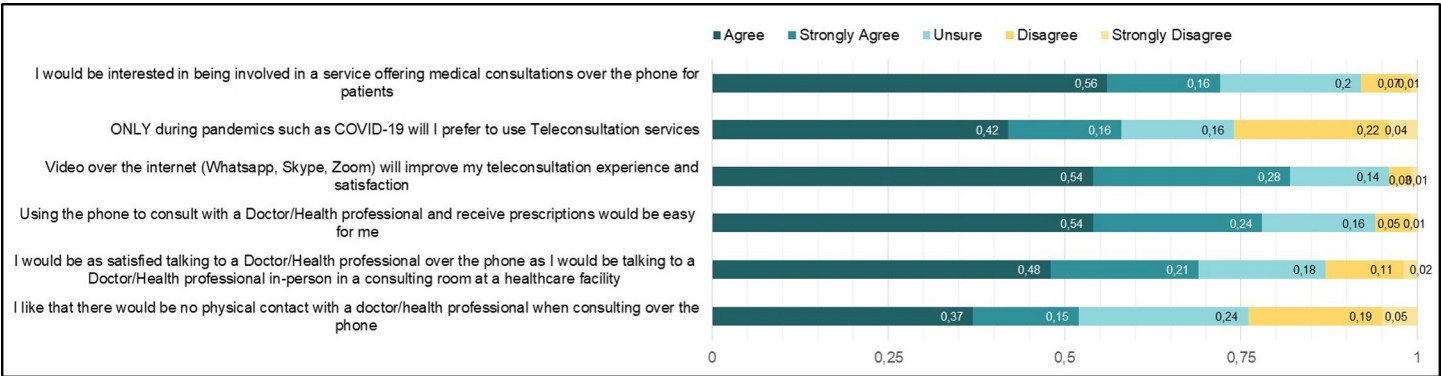

**Fig 3. Healthcare consumers overall perceptions on telemedicine use.**

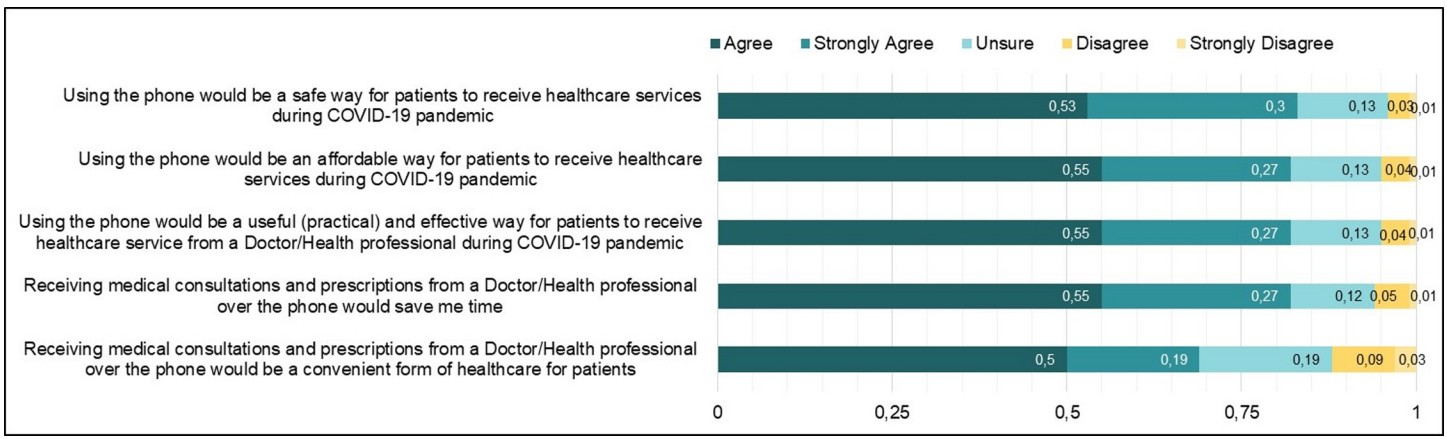

**Fig 4. Healthcare consumers perception on the benefits of telemedicine use.**

highlighted the need to consider telemedicine for healthcare delivery [17, 18]. This study explored healthcare providers' and consumers' perceptions of telemedicine utilization during Nigeria's ongoing COVID-19 pandemic. Overall, the results suggest that HCP and HCC are receptive to the use of telemedicine for health consultation and found it to be a useful and practical tool. Most HCP had experience with telemedicine (90%), and fewer HCC (63%) had used telemedicine but believed telemedicine should be expanded beyond the COVID-19 Pandemic.

## Comparison with previous work

Telemedicine service involving cellphone use appears to be the most popular form, with 84% of HCP and 50.6% of HCC reporting using this modality for teleconsultation or healthcare service. This is consistent with the study by Shittu et al. (2007) [19] in Lagos, Nigeria, where phones were the most popular form of telemedicine communication among study participants (85.9% of healthcare providers in the study).

Our study findings showed that HCP and HCC have favorable attitudes towards telemedicine use. This is congruent with previous studies in Nigeria [20] and other sub-Saharan African countries [21] that had assessed perceptions of telemedicine use. In addition, our study suggests that HCP and HCC perceived that telemedicine carried certain benefits for healthcare delivery. Most of the participants (both HCP and HCC) agreed that telemedicine could enhance the healthcare delivery experience by saving time and convenience. Specific to the healthcare delivery during the COVID-19 pandemic, most of the participants agreed that

**Table 3. Comparison of healthcare providers and health care consumers overall perceptions on telemedicine use.**

| Perceptions of Telemedicine Use | Healthcare Providers | Healthcare Consumers | P-Value |
|---|---|---|---|
| Agreed that they liked that there would be no physical contact with receiving or delivering health service when consulting over the phone | 34 (21.5) | 708 (51.5) | <0.001 |
| Agreed that using the phone to consult will be easy for delivering or receiving health service | 67 (42.4) | 1085 (78.9) | <0.001 |
| Agreed that video over the internet will improve teleconsultation experience and satisfaction | 98 (62.0) | 1133 (82.4) | <0.001 |
| Agreed that they would be as satisfied receiving or delivering service over the phone as they would in a consulting room at a healthcare facility | 39 (24.7) | 957 (69.6) | <0.001 |
| Agreed that they prefer to use teleconsultation services ONLY during pandemics | 86 (54.4) | 791 (57.9) | 0.497 |
| Agreed that they would be interested in being involved in a service offering medical consultations over the phone | 71 (44.9) | 997 (72.5) | <0.001 |

**Table 4. Comparison of healthcare providers and healthcare consumers perceptions on benefits of telemedicine use.**

| Benefits of Telemedicine Use | Healthcare Providers | Healthcare Consumers | P-Value |
|---|---|---|---|
| Agreed that using phone would be a useful (practical) and effective way to deliver or receive healthcare service during COVID-19 Pandemic | 84 (53.2) | 1135 (82.5) | <0.001 |
| Agreed that using the phone would be an affordable way to receive or deliver healthcare services during COVID-19 Pandemic | 96 (60.8) | 1135 (82.5) | <0.001 |
| Agreed that using the phone would be a safe way to deliver or to receive healthcare services during COVID-19 Pandemic | 102 (64.6) | 1150 (83.6) | <0.001 |
| Agreed that delivering and receiving medical consultations and prescriptions would be a convenient form of healthcare delivery | 64 (40.5) | 954 (62.2) | <0.001 |
| Agreed that delivering and receiving medical consultations and prescriptions over the phone would save me time | 94 (59.5) | 1127 (82.0) | <0.001 |

telemedicine would be an affordable, safe, and effective way to deliver or receive healthcare services. These benefits are similar to those in existing literature [22]. Notably, there were statistically significant differences in the proportions of HCP and HCC that agreed on the benefits of telemedicine use for healthcare service delivery. This difference in proportion could be attributed to differences between HCP and HCC overall knowledge of telehealth services. Ideally, HCP are more likely to be well informed about the challenges associated with the implementation of teleconsultation services in Nigeria more than the HCC, hence being more reluctant to acknowledge the uptake and benefits of these services. However, this difference in proportion could also be attributed to the difference in sample size between the two groups and might represent a confounding factor in this study.

Regarding participants' views on the cost of telemedicine, there were mixed responses from HCP and HCC, with the majority from each group indicating the cost should be less than in-person healthcare service. Although this study did not assess the influence of cost on telemedicine uptake, previous studies have reported that HCP are less likely to use telemedicine services if they are not adequately compensated for their time and effort [23, 24]. Likewise, the affordability of telemedicine for HCC is a critical factor in telemedicine uptake [24].

Overall, findings from this study support the acceptability of telemedicine for healthcare service delivery and consultation. This underscores the need for future work to evaluate how the acceptability of telemedicine translates to the implementation and adoption of healthcare systems. A recent review highlights technological, organizational, legal and regulatory, individual, financial, and cultural aspects as key barriers to the successful implementation of telemedicine in sub-Saharan Africa [24]. Hence concerted efforts are needed to address these barriers to facilitate the adoption of telemedicine for healthcare delivery during and beyond the COVID-19 pandemic.

## Strengths and limitations

The strengths of this study include, to the best of our knowledge, this is one of the first studies to explore the acceptability, perceptions of telemedicine among Nigerian HCP and the general public. This information is critical as we make sense of healthcare service delivery during the COVID-19 Pandemic and post-pandemic. Our study has some limitations worth noting. First, it is subject to selection bias due to the convenience sampling and recruitment approach. This means that the survey was limited to individuals who have smartphone devices and were technologically savvy to complete online surveys. This excludes the perspectives of individuals without access to the internet.

Therefore, findings from this study should be considered exploratory and may not be generalizable to the HCP and the public in Nigeria. However, considering the ongoing COVID-19 pandemic, this approach was the safest strategy and had the potential for a wider reach. Secondly, this study was also limited by the study's cross-sectional design, which precludes the investigation of causal relationships. Particularly, follow-up research, using a longitudinal design, could determine to what extent specific telehealth applications continue and evolve over time and after the Pandemic, assessing their perceived benefits and challenges for healthcare provision in the long term. Thirdly, all data was collected using an online survey. Notwithstanding the limitations, the results of this study provide information regarding providers' and individuals' perceptions on the use of telemedicine due to the COVID-19 pandemic. This can inform relevant public health response. Other variables of interest to consider in future studies include additional information on the challenges and nuanced benefits. Despite these limitations, this study adds new and important information to the literature on perception, acceptability, and uptake of telemedicine in Nigeria.

## Conclusions

Technology is playing an increasingly important role in enhancing patient healthcare access and delivery, particularly with the emergence of the ongoing COVID-19 pandemic. Telemedicine has been shown to aptly complement face-to-face hospital visits in ensuring effective triaging in hospitals and providing adequate healthcare delivery to patients regardless of geographical and physical barriers. In this study, most participants, HCP, and HCC expressed positive views towards telemedicine consultations. These results support telemedicine use for the provision of healthcare services during the COVID-19 pandemic. Future studies can investigate implementation strategies for the adoption of telemedicine and how perceptions evolve post COVID-19 pandemic.

## Supporting information

**S1 Survey. HCP questionnaire.**
(DOCX)

**S2 Survey. HCC questionnaire.**
(DOCX)

**S1 Checklist. PLOS questionnaire on inclusivity in global research.**
(DOCX)

**S1 Table. Results-association between healthcare providers demographic characteristics and perceptions on telemedicine use.**
(DOCX)

**S2 Table. Results-association between healthcare providers demographic characteristics and perceptions on benefits of telemedicine use.**
(DOCX)

**S3 Table. Results-association between healthcare consumers demographic characteristics and perceptions on telemedicine use.**
(DOCX)

**S4 Table. Results-association between healthcare consumers demographic characteristics and perceptions on benefits of telemedicine use.**
(DOCX)

## Acknowledgments

We thank all healthcare providers and healthcare consumers who participated in the study and the **Reunlearn360 2020 Mentee cohort** (Kikelomo Adetola Omotoye, Ebuka Samuel Anyaso, Fatima Zara Abdurahaman and Zainab Olamide Olaniyan) who helped in the survey dissemination.

## Author Contributions

**Conceptualization:** Ifeanyi Jude Ezeonwumelu.

**Data curation:** Ucheoma Nwaozuru.

**Formal analysis:** Ifeanyi Jude Ezeonwumelu, Ucheoma Nwaozuru.

**Investigation:** Ifeanyi Jude Ezeonwumelu, Ifeanyi Jude Obijiaku, Chukwudi Martin Ogbueche, Ucheoma Nwaozuru.

**Methodology:** Ifeanyi Jude Ezeonwumelu, Ifeanyi Jude Obijiaku, Chukwudi Martin Ogbueche, Ucheoma Nwaozuru.

**Project administration:** Ifeanyi Jude Ezeonwumelu.

**Supervision:** Ifeanyi Jude Ezeonwumelu, Ucheoma Nwaozuru.

**Validation:** Ifeanyi Jude Ezeonwumelu, Ucheoma Nwaozuru.

**Writing – original draft:** Ifeanyi Jude Ezeonwumelu, Ucheoma Nwaozuru.

**Writing – review & editing:** Ifeanyi Jude Ezeonwumelu, Ucheoma Nwaozuru.

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
