## [Decision Letter · Decision Letter 0]

17 Dec 2021

PGPH-D-21-00917

Healthcare provider-to-Patient perspectives on the uptake of Teleconsultation services in the Nigerian healthcare system in the COVID19 pandemic era

Dear Dr. EZEONWUMELU,

Thank you for submitting your manuscript to PLOS Global Public Health. After careful consideration, we feel that it has merit but does not fully meet PLOS Global Public Health’s publication criteria as it currently stands. Therefore, we invite you to submit a revised version of the manuscript that addresses the points raised during the review process.

We look forward to receiving your revised manuscript.

Kind regards,

**Joao** Tiago da Silva **Botelho**

Academic Editor

Journal Requirements:

1. Please include a complete copy of PLOS’ questionnaire on inclusivity in global research in your revised manuscript. Our policy for research in this area aims to improve transparency in the reporting of research performed outside of researchers’ own country or community. The policy applies to researchers who have travelled to a different country to conduct research, research with Indigenous populations or their lands, and research on cultural artefacts. The questionnaire can also be requested at the journal’s discretion for any other submissions, even if these conditions are not met.  Please find more information on the policy and a link to download a blank copy of the questionnaire here: https://journals.plos.org/plosone/s/best-practices-in-research-reporting. Please upload a completed version of your questionnaire as Supporting Information when you resubmit your manuscript.

2. We ask that a manuscript source file is provided at Revision. Please upload your manuscript file as a .doc, .docx, .rtf or .tex. If you are providing a .tex file, please upload it under the item type ‘LaTeX Source File’ and leave your .pdf version as the item type ‘Manuscript’.

3. Please update the completed 'Competing Interests' statement, including any COIs declared by your co-authors. If you have no competing interests to declare, please state "The authors have declared that no competing interests exist". Otherwise please declare all competing interests beginning with the statement "I have read the journal's policy and the authors of this manuscript have the following competing interests:"

4. Please amend your]detailed Financial Disclosure statement. This is published with the article, therefore should be completed in full sentences and contain the exact wording you wish to be published.

i) Please include all sources of funding (financial or material support) for your study. List the grants (with grant number) or organizations (with url) that supported your study, including funding received from your institution. 

ii). State the initials, alongside each funding source, of each author to receive each grant.

iii). State what role the funders took in the study. If the funders had no role in your study, please state: “The funders had no role in study design, data collection and analysis, decision to publish, or preparation of the manuscript.”

iv). If any authors received a salary from any of your funders, please state which authors and which funders.

Reviewers' comments:

Reviewer's Responses to Questions

**Comments to the Author**

1. Does this manuscript meet PLOS Global Public Health’s publication criteria? Is the manuscript technically sound, and do the data support the conclusions? The manuscript must describe methodologically and ethically rigorous research with conclusions that are appropriately drawn based on the data presented.

Reviewer #1: Yes

Reviewer #2: Partly

2. Has the statistical analysis been performed appropriately and rigorously?

Reviewer #1: Yes

Reviewer #2: Yes

3. Have the authors made all data underlying the findings in their manuscript fully available (please refer to the Data Availability Statement at the start of the manuscript PDF file)?

Reviewer #1: Yes

Reviewer #2: No

4. Is the manuscript presented in an intelligible fashion and written in standard English?

Reviewer #1: Yes

Reviewer #2: Yes

5. Review Comments to the Author

Reviewer #1: The authors have done a good job in this manuscript. The data analysis and presentation is excellent, and also the conclusion is well supported with the data in the results. However, there are some minor corrections required in the general text as follows:

Lines 152-153: delete “Healthcare providers” and use only its abbreviation because it was already abbreviated in Lines 112-113.

Lines 179-180: delete “healthcare providers” (line-179) and “healthcare consumers” (line-180) and use only their abbreviations.

Line 182: long form of UBTH is important to be inserted followed by UBTH in brackets, i.e., University of Benin Teaching Hospital (UBTH). Then, in subsequent areas where UBTH is written in long form, you can replace it with its abbreviation.

There is a need for consistency in the use of abbreviation for the Coronavirus disease of 2019. In most lines in the Introduction part, authors have used “Covid-19” but in other manuscript sections (including in the abstract), they have used COVID-19. For example, in Lines 185, 241, 279, 282, 286, etc., they used “COVID-19”.

Lines 380-381: delete “healthcare providers” (line-380) and “healthcare consumers” (line-381) and use only their abbreviations.

Line 414: “responses from healthcare providers and healthcare consumers,”, the authors need to use the abbreviations for HCP and HCC instead of their long-form.

Line 417: “telemedicine uptake, previous studies have reported that healthcare providers” – use HCP instead of its long form.

Line 420: “telemedicine for healthcare consumers is a critical factor in telemedicine uptake” – use HCC instead of its long form.

Line 434: “Nigerian healthcare providers and the general public. This information is critical” – replace “healthcare providers” with HCP after the word “Nigerian”.

Line 442: “be generalizable to the healthcare providers and the public in Nigeria. However,” – replace “healthcare provider” with HCP.

Lines 450-452: “using an online survey. Notwithstanding the limitations, the results of this study provide information providers' and individuals' perceptions on the use of telemedicine due to the coronavirus pandemic.” I SUGGEST THAT THE AUTHORS NEED TO REPHRASE IT TO READ AS FOLLOWS: ““using an online survey. Notwithstanding the limitations, the results of this study provide information REGARDING providers' and individuals' perceptions on the use of telemedicine due to the (((DELETE THIS coronavirus and insert))) COVID-19 pandemic.”

Lines 464-465: “barriers. In this study, most participants, healthcare providers, and healthcare consumers expressed positive…….” – replace “healthcare providers, and healthcare consumers” with “HCP and HCC”.

Reviewer #2: The author of the paper has done good work. The issues identified are:-

1. Study population is not clear. How many health care providers were sampled and how they identified, also to the healthcare consumers. how was their conduct obtained e.g. mobile number or how did participants both HCP and

3. How easy was healthcare consumers getting health providers conducts like phones and how was the consumers able to authenticate the numbers. Consumers recruited to the social media used to even deliver google forms used?. What was the response rate? Was stratification of the samples

2. Only one question for healthcare consumers as per the chart.

6. PLOS authors have the option to publish the peer review history of their article (what does this mean?). If published, this will include your full peer review and any attached files.

**Do you want your identity to be public for this peer review?** For information about this choice, including consent withdrawal, please see our Privacy Policy.

Reviewer #1: **Yes: **Eliudi Saria Eliakimu

Reviewer #2: No

---

## [Decision Letter · Decision Letter 1]

19 Jan 2022

Healthcare provider-to-Patient perspectives on the uptake of Teleconsultation services in the Nigerian healthcare system during the COVID-19 pandemic era

PGPH-D-21-00917R1

Dear Dr. EZEONWUMELU,

We're pleased to inform you that your manuscript has been judged scientifically suitable for publication and will be formally accepted for publication once it meets all outstanding technical requirements.

Within one week, you'll receive an e-mail detailing the required amendments. When these have been addressed, you'll receive a formal acceptance letter and your manuscript will be scheduled for publication.

An invoice for payment will follow shortly after the formal acceptance. To ensure an efficient process, please log into Editorial Manager at https://www.editorialmanager.com/pgph/ click the 'Update My Information' link at the top of the page, and double check that your user information is up-to-date. If you have any billing related questions, please contact our Author Billing department directly at authorbilling@plos.org.

Kind regards,

Joao Tiago da Silva Botelho

Academic Editor

Additional Editor Comments (optional):

Reviewers' comments:

Reviewer's Responses to Questions

**Comments to the Author**

1. If the authors have adequately addressed your comments raised in a previous round of review and you feel that this manuscript is now acceptable for publication, you may indicate that here to bypass the “Comments to the Author” section, enter your conflict of interest statement in the “Confidential to Editor” section, and submit your "Accept" recommendation.

Reviewer #1: All comments have been addressed

2. Does this manuscript meet PLOS Global Public Health’s publication criteria? Is the manuscript technically sound, and do the data support the conclusions? The manuscript must describe methodologically and ethically rigorous research with conclusions that are appropriately drawn based on the data presented.

Reviewer #1: Yes

3. Has the statistical analysis been performed appropriately and rigorously?

Reviewer #1: Yes

4. Have the authors made all data underlying the findings in their manuscript fully available (please refer to the Data Availability Statement at the start of the manuscript PDF file)?

Reviewer #1: Yes

5. Is the manuscript presented in an intelligible fashion and written in standard English?

Reviewer #1: Yes

6. Review Comments to the Author

Reviewer #1: (No Response)

7. PLOS authors have the option to publish the peer review history of their article (what does this mean?). If published, this will include your full peer review and any attached files.

**Do you want your identity to be public for this peer review?** For information about this choice, including consent withdrawal, please see our Privacy Policy.

Reviewer #1: **Yes: **Eliudi Saria Eliakimu
